# Brain-wide cellular resolution imaging of Cre transgenic zebrafish lines for functional circuit-mapping

Kathryn M Tabor[1], Gregory D Marquart[1,2], Christopher Hurt[1,3], Trevor S Smith[1], Alexandra K Geoca[1], Ashwin A Bhandiwad[1], Abhignya Subedi[4], Jennifer L Sinclair[1], Hannah M Rose[1], Nicholas F Polys[3], Harold A Burgess[1]*

[1]Division of Developmental Biology, Eunice Kennedy Shriver National Institute of Child Health and Human Development, Bethesda, United States; [2]Neuroscience and Cognitive Science Program, University of Maryland, College Park, United States; [3]Advanced Research Computing, Department of Computer Science, Virginia Polytechnic Institute and State University, Blacksburg, United States; [4]Postdoctoral Research Associate Training Program, National Institute of General Medical Sciences, Bethesda, United States

**Abstract** Decoding the functional connectivity of the nervous system is facilitated by transgenic methods that express a genetically encoded reporter or effector in specific neurons; however, most transgenic lines show broad spatiotemporal and cell-type expression. Increased specificity can be achieved using intersectional genetic methods which restrict reporter expression to cells that co-express multiple drivers, such as Gal4 and Cre. To facilitate intersectional targeting in zebrafish, we have generated more than 50 new Cre lines, and co-registered brain expression images with the Zebrafish Brain Browser, a cellular resolution atlas of 264 transgenic lines. Lines labeling neurons of interest can be identified using a web-browser to perform a 3D spatial search (zbbrowser.com). This resource facilitates the design of intersectional genetic experiments and will advance a wide range of precision circuit-mapping studies.
DOI: https://doi.org/10.7554/eLife.42687.001

*For correspondence:
burgessha@mail.nih.gov

Competing interests: The authors declare that no competing interests exist.

## Introduction

Elucidating the functional circuitry of the brain requires methods to visualize neuronal cell types and to reproducibly control and record activity from identified neurons. Genetically encoded reporters and effectors enable non-invasive manipulations in neurons but are limited by the precision with which they can be targeted. While gene regulatory elements are often exploited to direct transgene expression, very few transgenic lines strongly express reporter genes in a single cell type within a spatially restricted domain. Precise targeting of optogenetic reagents can be achieved using spatially restricted illumination in immobilized or optic-fiber implanted animals (*Aravanis et al., 2007*; *Arrenberg et al., 2009*; *Wyart et al., 2009*; *Zhu et al., 2012*). However, non-invasive genetic methods that confine expression to small groups of neurons enable analysis of behavior in freely moving animals. Such methods include intersectional genetic strategies, where reporter expression is controlled by multiple independently-expressed activators (*Dymecki et al., 2010*; *Gohl et al., 2011*).

In zebrafish, intersectional control of transgene expression has been achieved by combining the Gal4/UAS and Cre/lox systems (*Förster et al., 2017*; *Satou et al., 2013*; *Tabor et al., 2018*). Gal4-Cre intersectional systems take advantage of hundreds of existing Gal4 lines which are already widely used in zebrafish circuit neuroscience (*Asakawa et al., 2008*; *Bergeron et al., 2012*; *Scott et al., 2007*), but are limited by the relatively poor repertoire of existing Cre lines, and

**Table 1.** Summary of transgenic lines in ZBB2.
Total numbers of enhancer trap lines, and transgenic lines (made using promoter fragments from genes, or through BAC recombination), broken down by type: Gal4, Cre or fluorescent protein (FP). Right columns total the number of lines where genomic information driving the expression pattern is available. This information inherently exists for transgenic lines, and was derived through integration site mapping for enhancer trap lines.

|  | All lines (n = 264) | | | Mapped (n = 171) | | |
|---|---|---|---|---|---|---|
|  | Gal4 | Cre | FP | Gal4 | Cre | FP |
| Enhancer trap | 138 | 65 | 5 | 96 | 15 | 4 |
| Transgenic | 20 | 0 | 36 | 20 | 0 | 36 |
| Total | 158 | 65 | 41 | 116 | 15 | 40 |

DOI: https://doi.org/10.7554/eLife.42687.005

difficulty in identifying pairs of driver lines that co-express in neurons of interest. The first version of the Zebrafish Brain Browser (ZBB) atlas provided a partial solution to these problems by co-aligning high resolution image stacks of more than 100 transgenic lines, with an accuracy approaching the limit of biological variability (*Marquart et al., 2017*; *Marquart et al., 2015*). ZBB enabled users to conduct a 3D spatial search for lines with reporter expression in areas of interest and predict the area of intersection between Gal4 and Cre lines, aiding the design of intersectional genetic experiments. However, ZBB software required local installation and only included 9 Cre lines, limiting opportunities for intersectional targeting.

Here, we describe the ZBB2 atlas, which comprises whole-brain expression patterns for 264 transgenic lines, including 65 Cre lines, and 158 Gal4 lines. We generated more than 100 new enhancer trap lines that express Cre or Gal4 in diverse subsets of neurons, then registered a high resolution image of each to the original ZBB atlas. For 3D visualization of expression patterns and to facilitate spatial searches for experimentally useful lines, we now provide an online interface to the atlas. Collectively, ZBB2 labels almost all cellular regions within the brain and will facilitate the reproducible targeting of neuronal subsets for circuit-mapping studies.

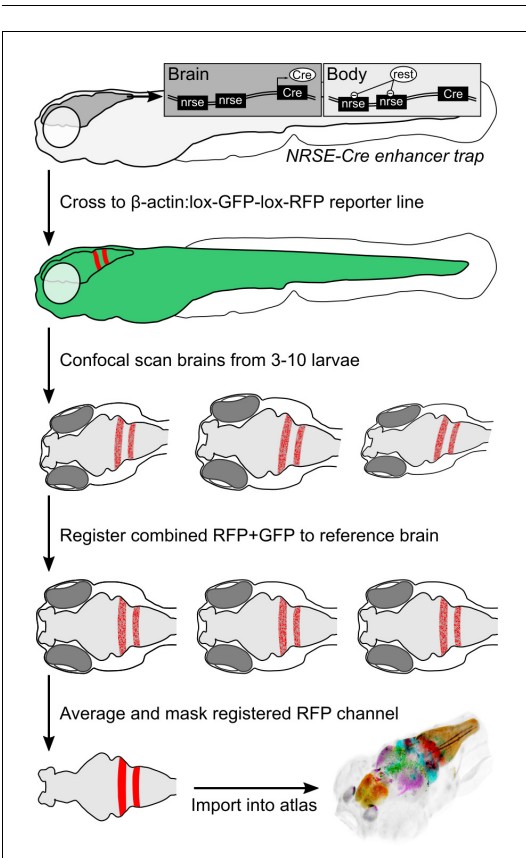

**Figure 1.** Procedure for imaging and co-registering new Cre lines. Inset schematics: neuronal-restrictive silencing element (NRSE) sites in the enhancer trap construct are targets for the REST protein, which suppresses Cre expression outside the brain.
DOI: https://doi.org/10.7554/eLife.42687.002

## Results

To accelerate discovery of functional circuits we generated a library of transgenic lines that express Cre in restricted patterns within the brain and built an online 3D atlas (*Figure 1*). New Cre lines were generated through an enhancer trap screen: we injected embryos with a Cre vector containing a basal promoter that includes a neuronal-restrictive silencing element to suppress expression outside the nervous system, and tol2 transposon arms for high efficiency transgenesis (*REx2-SCP1:BGi-Cre-2a-Cer*) (*Bergeron et al., 2012*; *Kawakami, 2007*;

*Marquart et al., 2015*). In injected G0 embryos, the reporter randomly integrates into the genome such that Cre expression is directed by local enhancer elements. To isolate lines with robust brain expression in relatively restricted domains, we visually screened progeny of G0 adults crossed to the *βactin:Switch* transgenic line that expresses red fluorescent protein (RFP) in cells with Cre (*Horstick et al., 2015*). We retained 52 new lines that express Cre in restricted brain regions at 6 days post-fertilization (dpf), the stage most frequently used for behavioral experiments and circuit-mapping. We then used a confocal microscope to scan brain-wide GFP and RFP fluorescence from *et-Cre, βactin:Switch* larvae at high resolution, aligned the merged signal to the ZBB *βactin:GFP* pattern and applied the resulting transformation matrix to the RFP signal alone. To obtain a representative image of Cre expression, we averaged registered brain scans from 3 to 10 larvae and masked expression outside the brain. Cre lines reported here for the first time are shown in *Figure 2*, and an overview of all 65 Cre lines that can be searched using ZBB2 is summarized in *Supplementary file 1*.

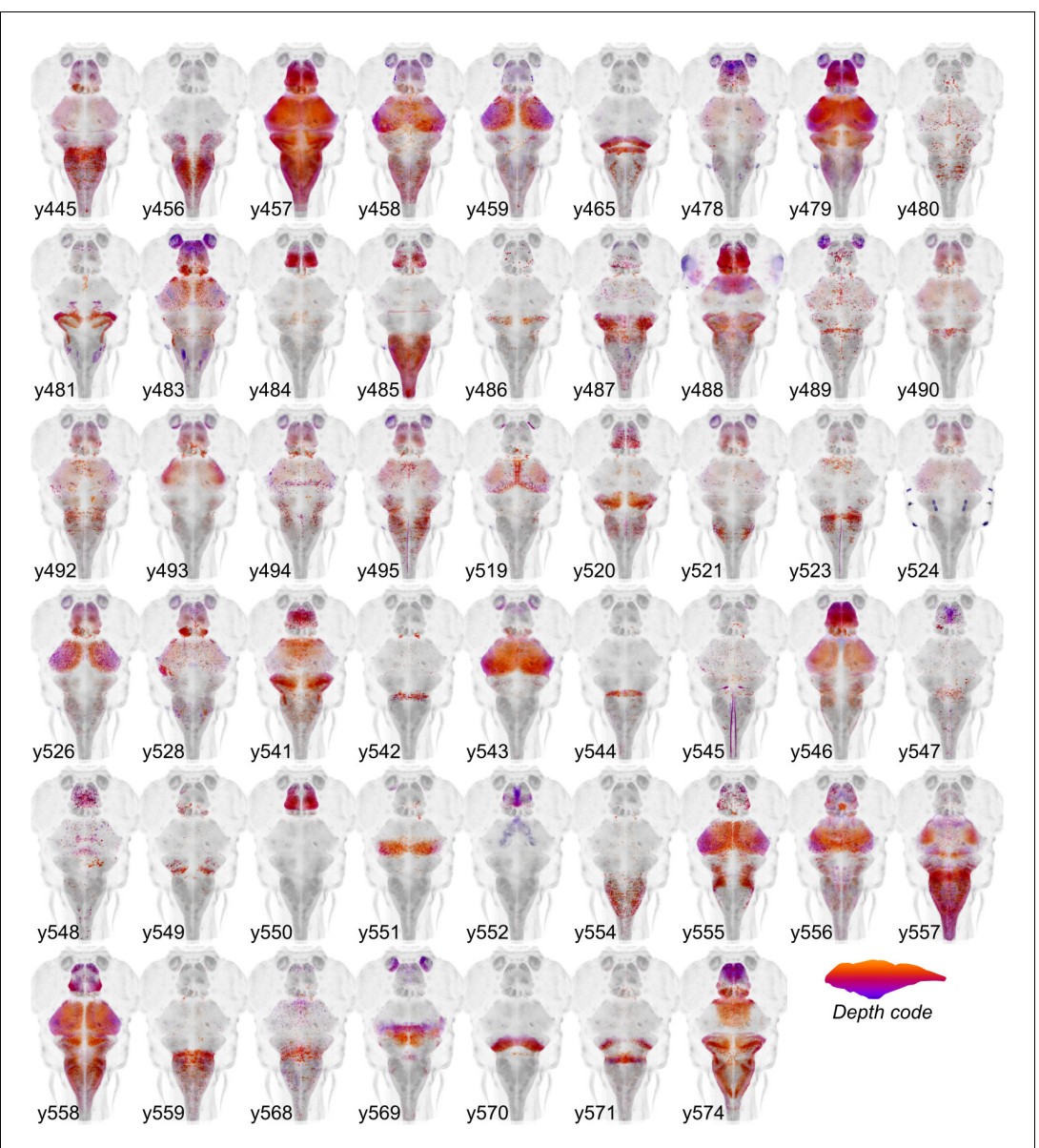

**Figure 2.** Cre enhancer trap lines. Horizontal maximum projection of 52 new Cre enhancer trap lines, with color indicating depth along the dorsal-ventral dimension (huC counter-label, grey).
DOI: https://doi.org/10.7554/eLife.42687.003

We previously isolated more than 200 Gal4 lines with expression in subregions of the brain; however, the original ZBB atlas represented only those that showed expression restricted to the nervous system. The subsequent development of a synthetic untranslated region (UTR.zb3) that suppresses non-neuronal expression has mitigated issues associated with Gal4 driving effector genes in non-neuronal tissues (*Marquart et al., 2015*). We therefore imaged 45 additional Gal4 lines in which robust brain expression is accompanied by expression in non-neural tissues. We visualized Gal4 expression using the *UAS:Kaede* transgenic line and registered patterns to ZBB by co-imaging *vglut2a:DsRed* expression. New Gal4 lines reported here are shown in *Figure 3*. We also aligned 20 high resolution brain scans of Gal4 lines performed by either the Dorsky or Baier laboratories by adapting a method for multi-channel registration of the Z-Brain and ZBB atlases (*Förster et al., 2017*; *Marquart et al., 2017*; *Otsuna et al., 2015*). In total, ZBB2 includes the spatial pattern of expression for 158 Gal4 lines. *Supplementary file 2* summarizes all Gal4 enhancer traps generated by our laboratory that can be searched in ZBB2. The relative expression of each transgenic line within 20 μm-side bins is reported in *Supplementary file 3*. In total, ZBB2 describes the expression pattern for 264 transgenic lines, more than doubling the number in the original atlas (*Table 1*).

Enhancer traps randomly integrate into the genome and it is not usually possible to determine the identity of the cells labeled by their spatial pattern of expression alone. However, cell-type information for enhancer trap lines can be inferred from co-localization with reporters whose expression

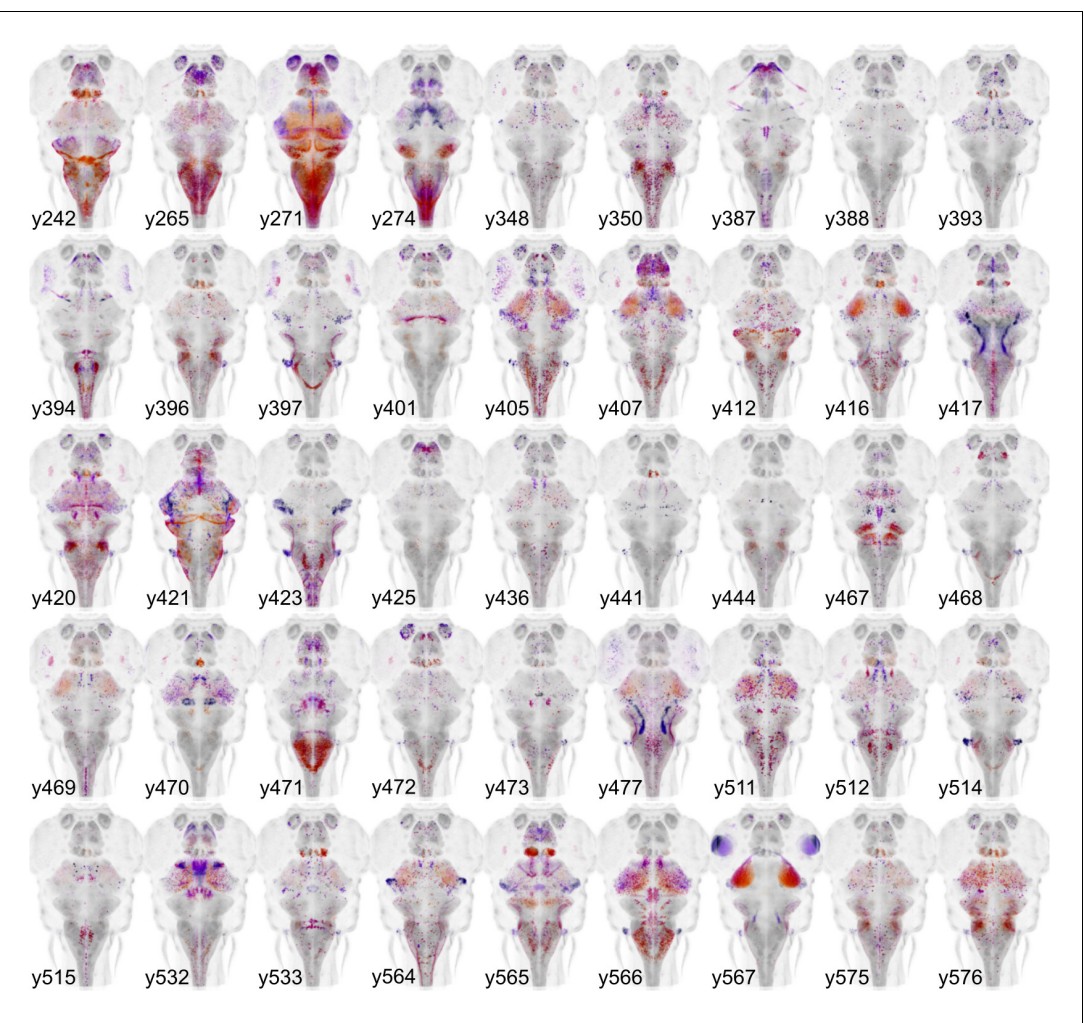

**Figure 3.** Gal4 enhancer trap lines. Horizontal maximum projection of 45 new Gal4 enhancer trap lines (depth coded; huC counter-label, grey).
DOI: https://doi.org/10.7554/eLife.42687.004

is directed by a defined promoter, or through integration into a bacterial artificial chromosome. ZBB2 contains expression data for 56 such transgenic lines, including reporters for most major neurotransmitters. The relative mean expression intensity for nine major cell-type markers within neuroanatomic. Additional cell-type information in enhancer trap lines may be revealed by integration site mapping, because enhancer traps often recapitulate, at least in part, the expression pattern of genes close to the site of transgene integration. We therefore developed a new method to efficiently map integration sites, using an oligonucleotide to hybridize with the enhancer trap tol2 arms and capture flanking genomic DNA fragments for sequencing (see Materials and methods for detail). We recovered the integration site for 55 Gal4 and Cre enhancer trap lines (detailed in ). Altogether 171 of the lines in ZBB2 either use a defined promoter, or have a known genomic integration site, providing molecular genetic information on cell-type identity (*Table 1*).

To assess how useful the lines represented in ZBB2 will be for circuit-mapping studies, we calculated the selectivity of each line. For this, we first estimated the volume of the brain that includes cell bodies — using transgenic markers of cell bodies and neuropil (*Figure 4A*) — then calculated the percent of the cell body volume labeled by each line. For Cre lines, median coverage of the cell body volume was 6% (range 0.1% to 48%, *Figure 4B,D*). As we estimate that there are ~92,000 neurons in the six dpf brain (see Materials and methods), this equates to a median of around 5500 neurons per line. In total, 96% of the cellular volume is labeled by at least one Cre line, with an average of 6 lines per voxel. However, Cre lines provide limited access to the midbrain tegmentum, posterior tuberculum and trigeminal ganglion (*Figure 4B*). Gal4 lines tend to have more restricted expression than Cre lines, with a median coverage of 1% (range 0.02% to 29%,~900 neurons per line). Collectively Gal4 lines label 91% of the cell body volume (*Figure 4C,D*). Salient areas that are not labeled include a rostro-dorsal domain of the optic tectum, the caudal lobe of the cerebellum, and a medial area within rhombomeres 3–4 of the medulla oblongata. Despite Gal4 lines having more restricted expression, few show tightly confined expression, highlighting the importance of intersectional approaches for precise targeting.

We envisaged using Cre lines to select experimentally useful subdomains of Gal4 expression patterns. To test this, we selected pairs of Cre and Gal4 lines with overlapping expression in spatially restricted regions, then analyzed reporter expression in triple transgenic Gal4/Cre/UAS:Switch larvae. As any given brain region likely contains multiple intermingled cell types, Gal4 and Cre lines

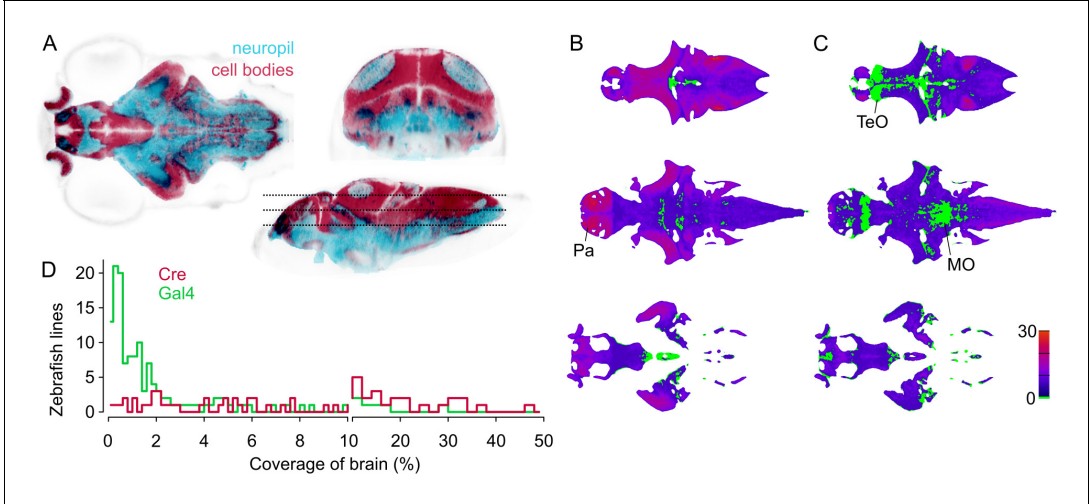

**Figure 4.** Spatial coverage of Cre and Gal4 enhancer trap lines. (**A**) Horizontal (left), coronal (top right), sagittal (bottom right) sections showing *huC:nls-mCar* (cell bodies, red) and *huC:Gal4,UAS:syp-RFP* (neuropil, cyan) from ZBB (huC counter-label, gray), illustrating the separation of cell bodies and neuropil in six dpf brains. (**B–C**) Horizontal sections (at the levels indicated in A) of heat-maps showing the number of *et-Cre* (**B**) and *et-Gal4* (**C**) lines that label each voxel within the cellular area of the brain (scale bar, right). Voxels that lack coverage indicated in green. Coverage for Cre lines is highest in the pallium (Pa). Gal4 lines conspicuously lack coverage in the anterior optic tectum (TeO) and in a medial zone of the medulla oblongata (MO). (**D**) Histogram of the cellular-region coverage for *et-Cre* (red) and *et-Gal4* (green) lines.
DOI: https://doi.org/10.7554/eLife.42687.006

that appear to overlap, may actually label distinct neurons. Nevertheless, intersectional patterns observed closely matched predicted intersects, labeling relatively small clusters of neurons (*Figure 5*). In general, intersectional expression in individual larvae is more restricted than predicted. In part, this may be because UAS transgenics are susceptible to silencing, reducing the extent of expression.

For the first release of the ZBB atlas, we provided downloadable *Brain Browser* software that allowed users to conduct a 3D spatial search for transgenic lines that label neurons within a specific Z-Brain defined neuroanatomic region (*Randlett et al., 2015*) or selected volume. For ZBB2, we have imported all the new lines into the original *Brain Browser*. However, recognizing that requiring a locally installed IDL runtime platform posed a limit to accessibility, we implemented an online version that can be accessed using a web-browser (http://zbbrowser.com). The online version includes key features of the original *Brain Browser*, including 3D spatial search, prediction of the area of intersectional expression between selected lines, partial/maximal/3D projections, information about the neuroanatomical identity of any selected voxel and ability to load user-generated image data (*Figure 6*). Additionally, the online version features an integrated virtual reality viewer for Google cardboard. We used X3DOM libraries to achieve rapid volume rendering and enable users to select data resolution that best matches their connection speed, so that the browser-based implementation remains highly responsive (*Arbelaiz et al., 2017b*; *Arbelaiz et al., 2017a*). Both local and web-based versions include hyper-links to PubMed, UCSC Genome Browser and Zfin, so that users can quickly retrieve publications describing each line, its integration site, and ordering information, respectively.

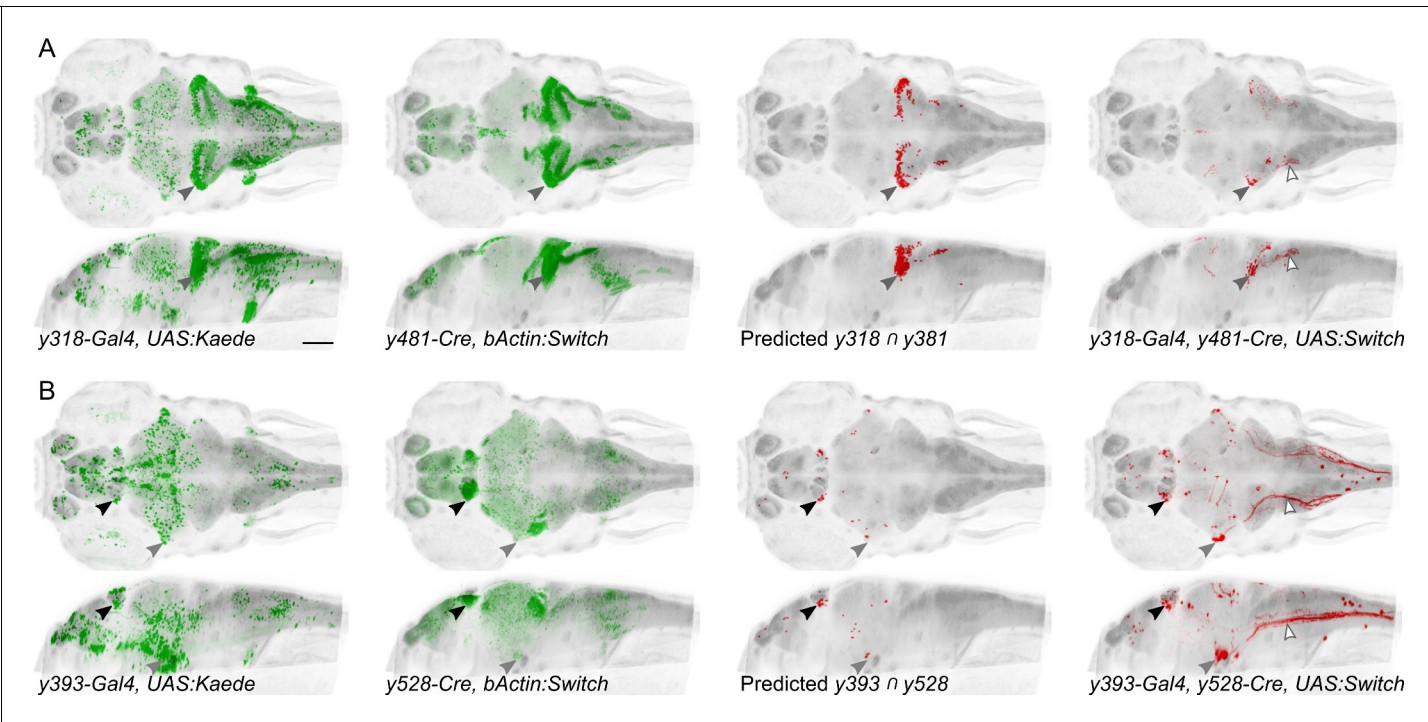

**Figure 5.** Spatially restricted reporter expression in Gal4/Cre intersectional domains. Maximum projections for et-Gal4 and et-Cre lines predicted to show co-expression in small domains of neurons, and of the resulting pattern of expression in a triple transgenic et-Gal4, et-Cre, UAS:KillSwitch larva. Scale bar 100 µm. (**A**) Horizontal (top) and sagittal (bottom) maximum projections for (left to right): *y318-Gal4* expression, *y481-Cre* expression, predicted reporter distribution in cells that co-express *y318* and *y481*, actual RFP expression in an individual *y318, y481, UAS:KillSwitch* larva. Closed arrowhead indicates the cerebellar eminentia granularis (Eg), which is labeled in both lines and the intersect. Open arrowhead indicates descending fiber tract from Eg neurons. (**B**) As for (**A**) with *y393-Gal4* and *y528-Cre* lines. Closed arrowheads indicate the lateral habenula (black) and trigeminal ganglion (grey). Open arrowhead indicates central projections from the trigeminal ganglion.
DOI: https://doi.org/10.7554/eLife.42687.007

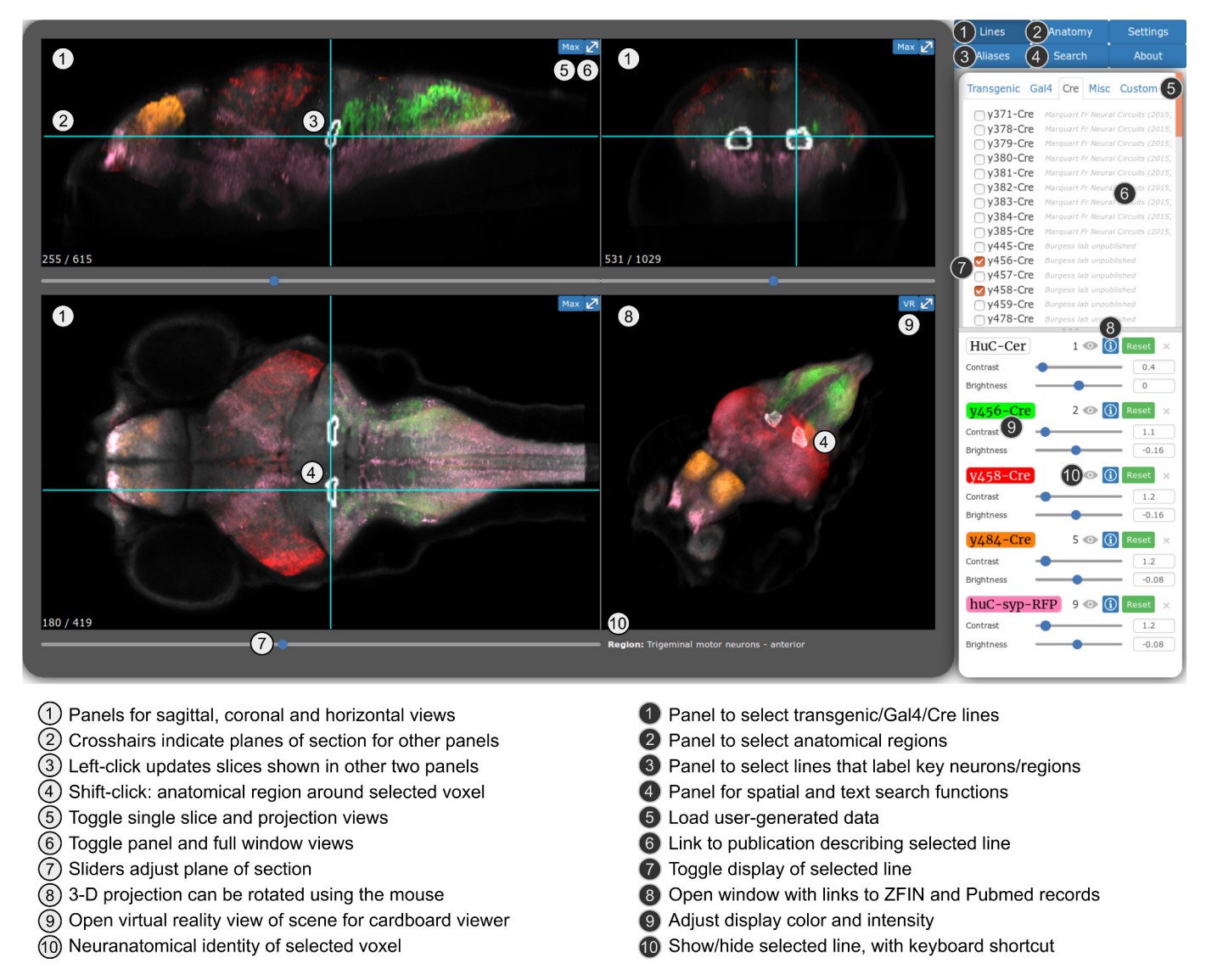

**Figure 6.** Web-browser based edition of the Zebrafish Brain Browser with key functions annotated. Panel size can be adjusted to best fit screen dimensions with the Settings menu.

DOI: https://doi.org/10.7554/eLife.42687.008

## Discussion

The ZBB2 atlas provides cellular resolution imaging data for 65 Cre lines, and 158 Gal4 lines, enabling small clusters of neurons to be genetically addressed through intersectional targeting. To aid in the design of such experiments, we have implemented a web-based interface that allows users to perform a spatial search for lines that express Cre or Gal4 in any selected 3D volume, and thereby identify transgenic lines for intersectional visualization or manipulation of selected neurons (*Tabor et al., 2018*). We anticipate that the ZBB2 lines will also facilitate structural mapping of the zebrafish brain because many strongly label discrete neuroanatomical entities.

In *Drosophila*, intersectional genetic targeting is often achieved using 'split' systems (Gal4, lexA and Q) in which the DNA-binding domain and transactivation domain of a transcription factor are separately expressed ensuring that activity is reconstituted only in neurons that express both protein domains during the same time interval (*Pfeiffer et al., 2010*; *Ting et al., 2011*; *Wei et al., 2012*). We did not pursue this approach because of the investment needed to create and maintain

numerous 'half' lines which can not be used alone. In contrast, Gal4 and Cre lines are independently useful and Cre lines can be used intersectionally with hundreds of existing zebrafish Gal4 lines. Several intersectional reporter lines have already been generated to visualize cells that co-express Cre and Gal4 (*Förster et al., 2017*; *Satou et al., 2013*). In addition, the *UAS:KillSwitch* line enables selective ablation of neurons that co-express Gal4 and Cre, and the *UAS:DoubleSwitch* line sparsely labels neurons within a Gal4/Cre co-expression domain for morphological reconstruction (*Tabor et al., 2018*).

New Gal4 and Cre lines described here were generated through enhancer trap screening, a high throughput method that facilitates the isolation of strongly expressing transgenic reporters in diverse anatomical regions and cell types. An alternative is to use CRISPR or TALEN based methods to insert transgenes into genes known to mark useful cell types or anatomical regions (*Auer et al., 2014*; *Kimura et al., 2014*). These methods are moderate throughput, and have the advantage of providing more reliable cell type information on the resulting transgene expression patterns. Limitations are that expression from an endogenous locus may not be sufficient to drive experimentally useful levels of Gal4 or Cre, and that an insertion may depress expression of the endogenous protein from the targeted allele. Given that many neurological disorders are associated with haploinsufficiency, reporter lines generated in this way may therefore perturb brain development or function. BAC transgenesis is a third method that has been widely used to make new Gal4 and Cre lines, that also exploits known gene expression patterns, while avoiding potentially disruptive insertions (*Förster et al., 2017*). However BAC transgenesis is low throughput, and as for targeted insertions, can not guarantee robust reporter expression. Thus future efforts at generating new transgenic reporters will likely be best served through enhancer traps and CRISPR-mediated transgene insertion.

A limitation of the Cre system is that insufficient expression may lead to stochastic activity at target loxP sites and consequent mosaic reporter expression (*Förster et al., 2017*; *Schmidt-Supprian and Rajewsky, 2007*). We therefore only retained lines with strong expression across multiple animals. Conversely Cre is toxic at high levels — in some cases, mosaic expression may be due to the death of a subset of cells (*Bersell et al., 2013*). While we cannot readily assess the toxicity of our lines, confounding effects can be experimentally addressed with the proper controls. Our enhancer trap Cre lines tend to have broader expression than Gal4 lines, likely because (1) Cre expression in progenitor cells labels all offspring, expanding the domain of expression, and (2) switch reporters remain active after transient Cre expression, whereas Gal4 must be continually expressed to drive UAS reporters. Occasionally we have observed UAS:Switch expression in neurons outside the domain of Cre expression in the ZBB2 atlas. This may be because the 14xUAS-E1b promoter is stronger than the β-actin promoter present in the switch reporter used to build the atlas. Additionally, most brain regions contain multiple cell types with biological variability in their precise position. Thus predicted overlapping expression based on co-registered brain scans must be experimentally verified.

The ZBB2 atlas advances mapping of the zebrafish brain at single cell resolution by comprehensively describing the cellular-resolution pattern of brain expression for 264 transgenic lines and providing a user-friendly web-browser based interface for searching and visualizing the expression in each. New Cre and Gal4 enhancer trap lines are freely available and we anticipate will advance circuit-mapping studies by providing essential reagents for intersectional targeting of neurons. We expect that this database and the associated transgenic lines will drive exploration of structure/function relations in the vertebrate brain.

## Materials and methods

**Key resources table**

| Reagent type (species) or resource | Designation | Source or reference | Identifiers | Additional information |
|---|---|---|---|---|
| Genetic reagent (*D. rerio*) | y234 | PMID:23100441 | ZFIN:ZDB -ALT-121114–10 | ZFIN symbol:y234Et; Et(SCP1:Gal4ff)y234 |

*Continued on next page*

*Continued*

| Reagent type (species) or resource | Designation | Source or reference | Identifiers | Additional information |
|---|---|---|---|---|
| Genetic reagent (*D. rerio*) | y236 | PMID:23293587 | ZFIN:ZDB-ALT-130214–2 | ZFIN symbol:y236Et; Et(REx2-cfos:kGal4ff)y236 |
| Genetic reagent (*D. rerio*) | y237 | PMID:23293587 | ZFIN:ZDB-ALT-130214–3 | ZFIN symbol:y237Et; Et(REx2-cfos:kGal4ff)y237 |
| Genetic reagent (*D. rerio*) | y241 | PMID:24203884 | ZFIN:ZDB-ALT-131007–1 | ZFIN symbol:y241Et; Et(REx2-SCP1:kGal4ff)y241 |
| Genetic reagent (*D. rerio*) | y242 | This paper | ZFIN:ZDB-ALT-130214–5 | ZFIN symbol:y242Et; Et(REx2-SCP1:Gal4)y242 |
| Genetic reagent (*D. rerio*) | y244 | PMID:23293587 | ZFIN:ZDB-ALT-130214–7 | ZFIN symbol:y244Et; Et(REx2-SCP1:kGal4ff)y244 |
| Genetic reagent (*D. rerio*) | y252 | PMID:25224259 | ZFIN:ZDB-ALT-151117–1 | ZFIN symbol:y252Et; Et(REx2-SCP1:kGal4ff)y252 |
| Genetic reagent (*D. rerio*) | y264 | PMID:24848468 | ZFIN:ZDB-ALT-141111–2 | ZFIN symbol:y264Et; Et(SCP1:Gal4ff)y264 |
| Genetic reagent (*D. rerio*) | y265 | This paper | ZFIN:ZDB-ALT-180717–10 | ZFIN symbol:y265Et; Et(SCP1:Gal4ff)y265 |
| Genetic reagent (*D. rerio*) | y269 | PMID:24848468 | ZFIN:ZDB-ALT-141111–3 | ZFIN symbol:y269Et; Et(REx2-cfos:kGal4ff)y269 |
| Genetic reagent (*D. rerio*) | y270 | PMID:24848468 | ZFIN:ZDB-ALT-141111–4 | ZFIN symbol:y270Et; Et(REx2-cfos:kGal4ff)y270 |
| Genetic reagent (*D. rerio*) | y271 | PMID:25628360 | ZFIN:ZDB-ALT-150721–4 | ZFIN symbol:y271Et; Et(SCP1:kGal4ff)y271 |
| Genetic reagent (*D. rerio*) | y274 | This paper | ZFIN:ZDB-ALT-180717–11 | ZFIN symbol:y274Et; Et(SCP1:Gal4ff)y274 |
| Genetic reagent (*D. rerio*) | y348 | This paper | ZFIN:ZDB-ALT-180717–12 | ZFIN symbol:y348Et; Et(REx2-SCP1:kGal4ff)y348 |
| Genetic reagent (*D. rerio*) | y387 | This paper | ZFIN:ZDB-ALT-180717–14 | ZFIN symbol:y387Et; Et(tph2:Gal4ff)y387 |
| Genetic reagent (*D. rerio*) | y394 | This paper | ZFIN:ZDB-ALT-180717–16 | ZFIN symbol:y394Et; Et(cfos:Gal4ff)y394 |
| Genetic reagent (*D. rerio*) | y396 | PMID:26635538 | ZFIN:ZDB-ALT-170320–13 | ZFIN symbol:y396Et; Et(SCP1:Gal4ff)y396 |
| Genetic reagent (*D. rerio*) | y412 | This paper | ZFIN:ZDB-ALT-180717–20 | ZFIN symbol:y412Et; Et(cfos:Gal4ff)y412 |
| Genetic reagent (*D. rerio*) | y416 | This paper | ZFIN:ZDB-ALT-180717–21 | ZFIN symbol:y416Et; Et(cfos:Gal4ff)y416 |
| Genetic reagent (*D. rerio*) | y420 | PMID:26635538 | ZFIN:ZDB-ALT-180717–23 | ZFIN symbol:y420Et; Et(REx2-cfos:Gal4ff)y420 |
| Genetic reagent (*D. rerio*) | y421 | This paper | ZFIN:ZDB-ALT-180717–24 | ZFIN symbol:y421Et; Et(REx2-cfos:Gal4)y421 |
| Genetic reagent (*D. rerio*) | y433 | This paper | ZFIN:ZDB-ALT-151125–10 | ZFIN symbol:y433Et; Et(cfos:kGal4ff)y433 |
| Genetic reagent (*D. rerio*) | y436 | This paper | ZFIN:ZDB-ALT-151125–13 | ZFIN symbol:y436Et; Et(cfos:Gal4ff)y436 |
| Genetic reagent (*D. rerio*) | y441 | This paper | ZFIN:ZDB-ALT-180717–27 | ZFIN symbol:y441Et; Et(cfos:Gal4ff)y441 |
| Genetic reagent (*D. rerio*) | y444 | This paper | ZFIN:ZDB-ALT-180717–28 | ZFIN symbol:y444Et; Et(SCP1:Gal4ff)y444 |
| Genetic reagent (*D. rerio*) | y445 | This paper | ZFIN:ZDB-ALT-180717–29 | ZFIN symbol:y445Et; Et(REx2-SCP1:BGi-Cre-2a-Cer.zf3)y445 |

*Continued on next page*

Continued

| Reagent type (species) or resource | Designation | Source or reference | Identifiers | Additional information |
|---|---|---|---|---|
| Genetic reagent (*D. rerio*) | y456 | This paper | ZFIN:ZDB-ALT-180717–30 | ZFIN symbol:y456Et; Et(REx2-SCP1:BGi-Cre-2a-Cer.zf3)y456 |
| Genetic reagent (*D. rerio*) | y457 | This paper | ZFIN:ZDB-ALT-180717–31 | ZFIN symbol:y457Et; Et(REx2-SCP1:BGi-Cre-2a-Cer.zf3)y457 |
| Genetic reagent (*D. rerio*) | y458 | This paper | ZFIN:ZDB-ALT-180717–32 | ZFIN symbol:y458Et; Et(REx2-SCP1:BGi-Cre-2a-Cer.zf3)y458 |
| Genetic reagent (*D. rerio*) | y459 | This paper | ZFIN:ZDB-ALT-180717–33 | ZFIN symbol:y459Et; Et(REx2-SCP1:BGi-Cre-2a-Cer.zf3)y459 |
| Genetic reagent (*D. rerio*) | y465 | This paper | ZFIN:ZDB-ALT-180717–34 | ZFIN symbol:y465Et; Et(REx2-SCP1:BGi-Cre-2a-Cer)y465 |
| Genetic reagent (*D. rerio*) | y467 | This paper | ZFIN:ZDB-ALT-180717–35 | ZFIN symbol:y467Et; Et(tph2:Gal4ff)y467 |
| Genetic reagent (*D. rerio*) | y468 | This paper | ZFIN:ZDB-ALT-180717–36 | ZFIN symbol:y468Et; Et(cfos:Gal4ff)y468 |
| Genetic reagent (*D. rerio*) | y469 | This paper | ZFIN:ZDB-ALT-180717–37 | ZFIN symbol:y469Et; Et(cfos:Gal4ff)y469 |
| Genetic reagent (*D. rerio*) | y470 | This paper | ZFIN:ZDB-ALT-180717–38 | ZFIN symbol:y470Et; Et(REx2-SCP1:Gal4ff)y470 |
| Genetic reagent (*D. rerio*) | y471 | This paper | ZFIN:ZDB-ALT-180717–39 | ZFIN symbol:y471Et; Et(REx2-SCP1:Gal4ff)y471 |
| Genetic reagent (*D. rerio*) | y472 | This paper | ZFIN:ZDB-ALT-180717–40 | ZFIN symbol:y472Et; Et(REx2-SCP1:Gal4ff)y472 |
| Genetic reagent (*D. rerio*) | y473 | This paper | ZFIN:ZDB-ALT-180717–41 | ZFIN symbol:y473Et; Et(cfos:Gal4ff)y473 |
| Genetic reagent (*D. rerio*) | y477 | This paper | ZFIN:ZDB-ALT-180717–43 | ZFIN symbol:y477Et; Et(cfos:Gal4ff)y477 |
| Genetic reagent (*D. rerio*) | y478 | This paper | ZFIN:ZDB-ALT-180717–44 | ZFIN symbol:y478Et; Et(attP-REx2-SCP1:BGi-Cre-2a-Cer-attP)y478 |
| Genetic reagent (*D. rerio*) | y479 | This paper | ZFIN:ZDB-ALT-180717–45 | ZFIN symbol:y479Et; Et(attP-REx2-SCP1:BGi-Cre-2a-Cer-attP)y479 |
| Genetic reagent (*D. rerio*) | y480 | This paper | ZFIN:ZDB-ALT-180717–46 | ZFIN symbol:y480Et; Et(attP-REx2-SCP1:BGi-Cre-2a-Cer-attP)y480 |
| Genetic reagent (*D. rerio*) | y481 | This paper | ZFIN:ZDB-ALT-180717–47 | ZFIN symbol:y481Et; Et(attP-REx2-SCP1:BGi-Cre-2a-Cer-attP)y481 |
| Genetic reagent (*D. rerio*) | y483 | This paper | ZFIN:ZDB-ALT-180717–48 | ZFIN symbol:y483Et; Et(attP-REx2-SCP1:BGi-Cre-2a-Cer-attP)y483 |
| Genetic reagent (*D. rerio*) | y484 | This paper | ZFIN:ZDB-ALT-180717–49 | ZFIN symbol:y484Et; Et(REx2-SCP1:BGi-Cre-2a-Cer)y484 |
| Genetic reagent (*D. rerio*) | y485 | This paper | ZFIN:ZDB-ALT-180717–50 | ZFIN symbol:y485Et; Et(REx2-SCP1:BGi-Cre-2a-Cer.zf3)y485 |
| Genetic reagent (*D. rerio*) | y486 | This paper | ZFIN:ZDB-ALT-180717–51 | ZFIN symbol:y486Et; Et(REx2-SCP1:BGi-Cre-2a-Cer.zf3)y486 |

*Continued on next page*

*Continued*

| Reagent type (species) or resource | Designation | Source or reference | Identifiers | Additional information |
|---|---|---|---|---|
| Genetic reagent (*D. rerio*) | y487 | This paper | ZFIN:ZDB-ALT-180717–52 | ZFIN symbol:y487Et; Et(REx2-SCP1:BGi-Cre-2a-Cer.zf3)y487 |
| Genetic reagent (*D. rerio*) | y488 | This paper | ZFIN:ZDB-ALT-180717–53 | ZFIN symbol:y488Et; Et(REx2-SCP1:BGi-Cre-2a-Cer.zf3)y488 |
| Genetic reagent (*D. rerio*) | y489 | This paper | ZFIN:ZDB-ALT-180717–54 | ZFIN symbol:y489Et; Et(REx2-SCP1:BGi-Cre-2a-Cer.zf3)y489 |
| Genetic reagent (*D. rerio*) | y490 | This paper | ZFIN:ZDB-ALT-180717–55 | ZFIN symbol:y490Et; Et(REx2-SCP1:BGi-Cre-2a-Cer.zf3)y490 |
| Genetic reagent (*D. rerio*) | y492 | This paper | ZFIN:ZDB-ALT-180717–56 | ZFIN symbol:y492Et; Et(REx2-SCP1:BGi-Cre-2a-Cer.zf3)y492 |
| Genetic reagent (*D. rerio*) | y493 | This paper | ZFIN:ZDB-ALT-180717–57 | ZFIN symbol:y493Et; Et(REx2-SCP1:BGi-Cre-2a-Cer.zf3)y493 |
| Genetic reagent (*D. rerio*) | y494 | This paper | ZFIN:ZDB-ALT-180717–58 | ZFIN symbol:y494Et; Et(REx2-SCP1:BGi-Cre-2a-Cer.zf3)y494 |
| Genetic reagent (*D. rerio*) | y495 | This paper | ZFIN:ZDB-ALT-180717–59 | ZFIN symbol:y495Et; Et(REx2-SCP1:BGi-Cre-2a-Cer.zf3)y495 |
| Genetic reagent (*D. rerio*) | y511 | This paper | ZFIN:ZDB-ALT-180717–60 | ZFIN symbol:y511Et; Et(cfos:Gal4ff)y511 |
| Genetic reagent (*D. rerio*) | y512 | This paper | ZFIN:ZDB-ALT-180717–61 | ZFIN symbol:y512Et; Et(SCP1:Gal4)y512 |
| Genetic reagent (*D. rerio*) | y514 | This paper | ZFIN:ZDB-ALT-180717–63 | ZFIN symbol:y514Et; Et(REx2-cfos:kGal4ff)y514 |
| Genetic reagent (*D. rerio*) | y515 | This paper | ZFIN:ZDB-ALT-180717–64 | ZFIN symbol:y515Et; Et(REx2-cfos:kGal4ff)y515 |
| Genetic reagent (*D. rerio*) | y519 | This paper | ZFIN:ZDB-ALT-180717–65 | ZFIN symbol:y519Et; Et(REx2-SCP1:BGi-Cre-2a-Cer)y519 |
| Genetic reagent (*D. rerio*) | y520 | This paper | ZFIN:ZDB-ALT-180717–66 | ZFIN symbol:y520Et; Et(REx2-SCP1:BGi-Cre-2a-Cer)y520 |
| Genetic reagent (*D. rerio*) | y521 | This paper | ZFIN:ZDB-ALT-180717–67 | ZFIN symbol:y521Et; Et(REx2-SCP1:BGi-Cre-2a-Cer.zf3)y521 |
| Genetic reagent (*D. rerio*) | y523 | This paper | ZFIN:ZDB-ALT-180717–68 | ZFIN symbol:y523Et; Et(REx2-SCP1:BGi-Cre-2a-Cer.zf3)y523 |
| Genetic reagent (*D. rerio*) | y524 | This paper | ZFIN:ZDB-ALT-180717–69 | ZFIN symbol:y524Et; Et(REx2-SCP1:BGi-Cre-2a-Cer)y524 |
| Genetic reagent (*D. rerio*) | y526 | This paper | ZFIN:ZDB-ALT-180717–70 | ZFIN symbol:y526Et; Et(REx2-SCP1:BGi-Cre-2a-Cer.zf3)y526 |
| Genetic reagent (*D. rerio*) | y528 | This paper | ZFIN:ZDB-ALT-180717–71 | ZFIN symbol:y528Et; Et(REx2-SCP1:BGi-Cre-2a-Cer.zf3)y528 |
| Genetic reagent (*D. rerio*) | y532 | This paper | ZFIN:ZDB-ALT-180717–72 | ZFIN symbol:y532Et; Et(cfos:Gal4ff)y532 |

*Continued on next page*

*Continued*

| Reagent type (species) or resource | Designation | Source or reference | Identifiers | Additional information |
|---|---|---|---|---|
| Genetic reagent (*D. rerio*) | y533 | This paper | ZFIN:ZDB-ALT-180717–73 | ZFIN symbol:y533Et; Et(SCP1:Gal4ff)y533 |
| Genetic reagent (*D. rerio*) | y541 | This paper | ZFIN:ZDB-ALT-180717–74 | ZFIN symbol:y541Et; Et(REx2-SCP1:BGi-Cre-2a-Cer.zf3)y541 |
| Genetic reagent (*D. rerio*) | y542 | This paper | ZFIN:ZDB-ALT-180717–75 | ZFIN symbol:y542Et; Et(REx2-SCP1:BGi-Cre-2a-Cer.zf3)y542 |
| Genetic reagent (*D. rerio*) | y543 | This paper | ZFIN:ZDB-ALT-180717–76 | ZFIN symbol:y543Et; Et(REx2-SCP1:BGi-Cre-2a-Cer.zf3)y543 |
| Genetic reagent (*D. rerio*) | y544 | This paper | ZFIN:ZDB-ALT-180717–77 | ZFIN symbol:y544Et; Et(REx2-SCP1:BGi-Cre-2a-Cer.zf3)y544 |
| Genetic reagent (*D. rerio*) | y545 | This paper | ZFIN:ZDB-ALT-180717–78 | ZFIN symbol:y545Et; Et(REx2-SCP1:BGi-Cre-2a-Cer.zf3)y545 |
| Genetic reagent (*D. rerio*) | y546 | This paper | ZFIN:ZDB-ALT-180717–79 | ZFIN symbol:y546Et; Et(REx2-SCP1:BGi-Cre-2a-Cer.zf3)y546 |
| Genetic reagent (*D. rerio*) | y547 | This paper | ZFIN:ZDB-ALT-180717–80 | ZFIN symbol:y547Et; Et(REx2-SCP1:BGi-Cre)y547 |
| Genetic reagent (*D. rerio*) | y548 | This paper | ZFIN:ZDB-ALT-180717–81 | ZFIN symbol:y548Et; Et(REx2-SCP1:BGi-Cre)y548 |
| Genetic reagent (*D. rerio*) | y549 | This paper | ZFIN:ZDB-ALT-180717–82 | ZFIN symbol:y549Et; Et(REx2-SCP1:BGi-Cre)y549 |
| Genetic reagent (*D. rerio*) | y550 | This paper | ZFIN:ZDB-ALT-180717–83 | ZFIN symbol:y550Et; Et(REx2-SCP1:BGi-Cre)y550 |
| Genetic reagent (*D. rerio*) | y551 | This paper | ZFIN:ZDB-ALT-180717–84 | ZFIN symbol:y551Et; Et(REx2-SCP1:BGi-Cre-2a-Cer.zf3)y551 |
| Genetic reagent (*D. rerio*) | y552 | This paper | ZFIN:ZDB-ALT-180717–85 | ZFIN symbol:y552Et; Et(REx2-SCP1:BGi-Cre-2a-Cer.zf3)y552 |
| Genetic reagent (*D. rerio*) | y554 | This paper | ZFIN:ZDB-ALT-180717–87 | ZFIN symbol:y554Et; Et(REx2-SCP1:BGi-Cre-2a-Cer.zf3)y554 |
| Genetic reagent (*D. rerio*) | y555 | This paper | ZFIN:ZDB-ALT-180717–88 | ZFIN symbol:y555Et; Et(REx2-SCP1:BGi-Cre)y555 |
| Genetic reagent (*D. rerio*) | y556 | This paper | ZFIN:ZDB-ALT-180717–89 | ZFIN symbol:y556Et; Et(REx2-SCP1:BGi-Cre)y556 |
| Genetic reagent (*D. rerio*) | y557 | This paper | ZFIN:ZDB-ALT-180717–90 | ZFIN symbol:y557Et; Et(REx2-SCP1:BGi-Cre)y557 |
| Genetic reagent (*D. rerio*) | y558 | This paper | ZFIN:ZDB-ALT-180717–91 | ZFIN symbol:y558Et; Et(REx2-SCP1:BGi-Cre-2a-Cer.zf3)y558 |
| Genetic reagent (*D. rerio*) | y559 | This paper | ZFIN:ZDB-ALT-180717–92 | ZFIN symbol:y559Et; Et(REx2-SCP1:BGi-Cre)y559 |
| Genetic reagent (*D. rerio*) | y564 | This paper | ZFIN:ZDB-ALT-180717–93 | ZFIN symbol:y564Et; Et(SCP1:Gal4ff)y564 |
| Genetic reagent (*D. rerio*) | y565 | This paper | ZFIN:ZDB-ALT-180717–94 | ZFIN symbol:y565Et; Et(REx2-SCP1:Gal4ff)y565 |
| Genetic reagent (*D. rerio*) | y566 | This paper | ZFIN:ZDB-ALT-180717–95 | ZFIN symbol:y566Et; Et(REx2-cfos:Gal4ff)y566 |

*Continued on next page*

*Continued*

| Reagent type (species) or resource | Designation | Source or reference | Identifiers | Additional information |
|---|---|---|---|---|
| Genetic reagent (*D. rerio*) | y567 | This paper | ZFIN:ZDB-ALT-180717–96 | ZFIN symbol:y567Et; Et(REx2-cfos:Gal4ff)y567 |
| Genetic reagent (*D. rerio*) | y568 | This paper | ZFIN:ZDB-ALT-180717–97 | ZFIN symbol:y568Et; Et(REx2-SCP1:BGi-Cre-2a-Cer.zf3)y568 |
| Genetic reagent (*D. rerio*) | y569 | This paper | ZFIN:ZDB-ALT-180717–98 | ZFIN symbol:y569Et; Et(REx2-SCP1:BGi-Cre)y569 |
| Genetic reagent (*D. rerio*) | y570 | This paper | ZFIN:ZDB-ALT-180717–99 | ZFIN symbol:y570Et; Et(REx2-SCP1:BGi-Cre)y570 |
| Genetic reagent (*D. rerio*) | y571 | This paper | ZFIN:ZDB-ALT-180717–100 | ZFIN symbol:y571Et; Et(R2R6-hoxa2-CNE-SCP1:BGi-Cre-2a-Cer)y571 |
| Genetic reagent (*D. rerio*) | y574 | This paper | ZFIN:ZDB-ALT-180717–101 | ZFIN symbol:y574Et; Et(REx2-SCP1:BGi-Cre)y574 |
| Genetic reagent (*D. rerio*) | y575 | This paper | ZFIN:ZDB-ALT-180717–102 | ZFIN symbol:y575Et; Et(cfos:Gal4)y575 |
| Genetic reagent (*D. rerio*) | y576 | This paper | ZFIN:ZDB-ALT-180717–103 | ZFIN symbol:y576Et; Et(cfos:Gal4)y576 |
| Genetic reagent (*D. rerio*) | βactin:Switch; y272 | PMID:25628360 | ZFIN:ZDB-ALT-150721–8 | |
| Genetic reagent (*D. rerio*) | vglut2a:DsRed; nns14 | PMID:22302816 | ZFIN:ZDB-ALT-110413–5 | |
| Genetic reagent (*D. rerio*) | UAS:KillSwitch; y518 | PMID:30078569 | ZFIN:ZDB-ALT-181218–5 | |
| Genetic reagent (*D. rerio*) | UAS:Kaede; s1999t | PMID:17335798 | ZFIN:ZDB-ALT-070314–1 | |
| Genetic reagent (*D. rerio*) | elavl3:h2b-GCaMP6; jf5 | PMID:25068735 | ZFIN:ZDB-ALT-141023–2 | |
| Recombinant DNA reagent | REx2-SCP1:BGi-Cre-2a-Cer (plasmid) | PMID:26635538 | ZFIN:ZDB-ETCONSTRCT-151102–1 | |
| Recombinant DNA reagent | REx2-SCP1:BGi-Cre | This paper | ZFIN:ZDB-ETCONSTRCT-180514–2 | Progenitors:REx2-SCP1:BGi-Cre-2a-Cer (plasmid) |
| Recombinant DNA reagent | REx2-SCP1:BGi-Cre-2a-Cer.zf3 (plasmid) | This paper | ZFIN:ZDB-ETCONSTRCT-180518–1 | Progenitors:REx2-SCP1:BGi-Cre-2a-Cer (plasmid) |
| Recombinant DNA reagent | attP-REx2-SCP1:BGi-Cre-2a-Cer-attP | This paper | ZFIN:ZDB-ETCONSTRCT-151102–2 | Progenitors:REx2-SCP1:BGi-Cre-2a-Cer (plasmid) |
| Recombinant DNA reagent | R2R6-hoxa2-CNE-SCP1:BGi-Cre-2a-Cer | This paper | ZFIN:ZDB-ETCONSTRCT-180514–1 | Progenitors:REx2-SCP1:BGi-Cre-2a-Cer (plasmid) |
| Sequence-based reagent | tol2-arrm pulldown oligonucleotide | This paper | | 5-CTCAAGTGAAAGTACAAGTACTTAGGGAAAATTTTACTCAATTAAAAGTAAAAGTATCTGGCTAGAATCTTACTTGAGTAAAAGTAAAAAAGTACTCCATTAAAATTGTACTTGAGTATT |
| Sequence-based reagent | tol2-arrm pulldown oligonucleotide | This paper | | 5-TGTAATTAAGTAAAAGTAAAAGTATTGATTTTTAATTGTACTCAAGTAAAGTAAAAATCCCCAAAAATAATACTTAAGTACAGTAATCAAGTAAAATTACTCAAGTACTTTACACCTCTG |

*Continued on next page*

*Continued*

| Reagent type (species) or resource | Designation | Source or reference | Identifiers | Additional information |
|---|---|---|---|---|
| Software | Advanced Normalization Tools | PMID:17659998 | | |
| Software | Zebrafish Brain Browser (desktop) | PMID:26635538 | | Download at https://science.nichd.nih.gov/confluence/display/burgess/Brain+Browser |
| Software | Zebrafish Brain Browser (online) | This paper | GitHub:BurgessLab/ZebrafishBrainBrowser | zbbrowser.com; Githhub hosts javascript code using X3DOM to render image files |
| Software | ImageJ | PMID:22930834 | | |
| Other | Confocal images for ZBB2 lines | This paper | Dryad:DOI: 10.5061/dryad.tk467n8 | Compressed archives containing 16-bit NIFTI format scans of individual larvae |

## Husbandry

Zebrafish (*Danio rerio*) were maintained on a Tubingen long fin strain background. Larval zebrafish were raised on 14/10 hr light/dark cycle at 28°C in E3h medium (5 mM NaCl, 0.17 mM KCl, 0.33 mM CaCl2, 0.33 mM MgSO4, 1.5 mM HEPES, pH 7.3) with 300 µM N-Phenylthiourea (PTU, Sigma) to suppress melanogenesis for imaging. Experiments were conducted with larvae at 6 days post fertilization (dpf), before sex differentiation. Experimental procedures were approved by the NICHD animal care and use committee.

## Zebrafish lines

Enhancer trap lines that express Cre (*et-Cre* lines) were initially isolated through enhancer trap screening using a tol2 vector containing a REx2-SCP1:BGi-Cre-2a-Cer cassette (286 adult fish screened) (*Marquart et al., 2015*). Although the fluorescent protein Cerulean is co-expressed with Cre in this vector, it was rarely strong enough to visualize directly, and we instead screened using the *βactin:Switch* transgenic line (*Horstick et al., 2015*). Thus subsequently, we removed the 2a-Cer cassette from the enhancer trap vector for generating new lines and injected a vector with a REx2-SCP1:BGi-Cre cassette (75 adult fish screened). At least 50 (usually over 100) offspring of injected fish were visually screened for RFP fluorescence from the *βactin:Switch (Tg(actb2:loxP-eGFP-loxP-ly-TagRFPT)y272)* reporter line (*Horstick et al., 2015*). As around 10% of injected animals transmitted more than a single expression pattern, likely reflecting several integration loci, we bred each line for multiple generations to isolate a single heritable transgene. Gal4 lines newly described here have been maintained for at least seven generations, in all cases with the UAS:Kaede reporter for visualizing expression. New Cre lines have been maintained for at least three generations, with the *βactin: Switch* reporter. Because we only retained lines with restricted areas of brain expression, we ultimately kept lines from around 20% of injected fish. For maintenance, Cre lines were crossed to fish heterozygous for the *βactin:Switch* transgene. Outcrossing to *βactin:Switch* was necessary because, as in other systems, leaky Cre expression recombines lox sites that are transmitted through the same gamete (*Schmidt-Supprian and Rajewsky, 2007*). Consequently, in clutches from *et-Cre;β actin:Switch* crossed to *βactin:Switch*, we discarded ~25% of embryos that showed ubiquitous RFP expression due to complete recombination of the Switch reporter in gametes also containing the Cre transgene. We also imaged previously described Cre lines with rhombomere-specific expression (*Tabor et al., 2018*). Gal4 enhancer trap lines were isolated as previously described (*Bergeron et al., 2012*). UAS reporter transgenes are susceptible to silencing leading to variegated expression. To minimize silencing, we raise single insertion UAS reporters with a broadly expressed Gal4 transgene. In each generation, we outcross to wildtype stock, and raise only double transgenic individuals with the brightest and most complete expression (typically around 20% of fluorescent protein positive embryos).

Other zebrafish lines in this study were: *UAS:Kaede (Tg(UAS-E1b:Kaede)s1999t* (*Davison et al., 2007*), *UAS:KillSwitch (Tg(14xUAS-E1b:BGi-lox-GFP-sv40-loxepNTR-TagRFPT)y518)* (*Tabor et al., 2018*), *y379-Cre* and *y484-Cre* (*Marquart et al., 2017*), *vglut2a:DsRed (TgBAC(slc17a6b:loxP-DsRed-loxP-GFP)nns9)* (*Satou et al., 2013*), *Tg(gata1:dsRed)sd2* (*Traver et al., 2003*), *Tg(−4.9sox10:EGFP)ba2* (*Wada et al., 2005*), *Tg(−8.4neurog1:GFP)sb1* (*Blader et al., 2003*), *Tg(kctd15a:GFP)y534* (*Heffer et al., 2017*), *Tg(pou4f3:gap43-GFP)s356t* (*Xiao et al., 2005*), *Et(−1.5hsp70l:Gal4-VP16)s1156t* and *Et(fos:Gal4-VP16)s1181t* (*Scott and Baier, 2009*), *TgBAC(neurod:EGFP)nl1* (*Obholzer et al., 2008*), *Tg(mnx1:GFP)ml2* (*Flanagan-Steet et al., 2005*), and *y271-Gal4* (*Horstick et al., 2015*). For counting neurons in the brain, we used *huC:h2b-GCaMP6 (Tg(elavl3:h2b-GCaMP6)jf5)* (*Vladimirov et al., 2014*), which has multiple transgene integrations, minimizing effects of variable expression and silencing.

## Brain imaging and processing

For imaging, six dpf larvae were embedded in 1.5–3.5% agarose in E3h and oriented dorsal to the objective. Each larval brain was scanned in two image stacks (anterior and posterior halves, 1 × 1×2 μm resolution) with an inverted Leica TCS-SP5 II confocal with a 25X, 0.95 NA objective, while adjusting laser power during scans to compensate for intensity loss with depth. Gal4 expression was visualized using *UAS:Kaede* and Cre expression using RFP expression from *βactin:Switch*. Color channels were usually acquired simultaneously and crosstalk removed in post-processing using a Leica dye separation algorithm. Substacks were connected using the pairwise stitching plugin in ImageJ (*Preibisch et al., 2009*; *Schneider et al., 2012*).

Image registration was performed using affine and diffeomorphic algorithms in ANTs (*Avants et al., 2011*) with parameters optimized for live embryonic zebrafish brain scans that produce alignments with an accuracy of approximately one cell diameter (8 μm) (*Marquart et al., 2017*). For registration, each image stack required a reference image previously registered to the ZBB coordinate system. Reference channels were *vglut2a:DsRed* for Gal4 lines and *βactin:Switch* GFP for Cre lines. Other transgenic lines and patterns were registered using either *vglut2a:dsRed* or *vglut2a:GFP* where appropriate. For Cre lines, we merged the *βactin:Switch* GFP and RFP signals into a combined pattern to provide a channel for registration. We then applied the resulting transformation matrix to the RFP channel alone. Next, we averaged registered brain images from at least three larvae per line, using the ANTs *AverageImages* command, to create a representative image of each line. Mean images were masked to remove expression outside the brain, except where inner ear hair cells or neuromasts were labeled. Next, we normalized intensity to saturate the top 0.01% of pixels, and downsampled to 8-bit to reduce file size and facilitate distribution. We manually defined fluorescent intensity thresholds for each line that best distinguished cellular expression from neuropil or background to facilitate spatial search for lines that express in selected cell populations. Mean images were also aligned to Z-Brain using a previously described bridging transformation matrix (*Marquart et al., 2017*).

Because registration using the *βactin:Switch* bridging channel proved more accurate than our previous bridging registration with *HuC:Cer*, we re-imaged and registered the Cre lines recovered in our pilot screen. Gal4 lines generated and imaged by the Dorsky lab (*Otsuna et al., 2015*) were registered using two channels: the nuclear counter-stain channel (TO-PRO−3) and immunolabeling for myosin heavy chain, aligned to *HuC:nls-mCar* and *tERK* in ZBB respectively. We also used multichannel registration to align brain scans performed by the Baier lab (*Förster et al., 2017*), taking advantage of three expression patterns present in both datasets: *vglut2a:dsRed*, *isl2b:GFP* and *gad1b:GFP*.

## Integration site mapping

To efficiently map enhancer trap integration sites we extracted genomic DNA from embryos from each line (Qiagen DNeasy Blood and Tissue Kit) and generated a barcoded library. We hybridized the library to biotinylated 120 bp primers (IDT ultramers) designed against the tol2 sequence arms and enriched for genomic integration sites using avidin-pulldown. Enriched libraries were combined into 15 pools such that each pool contained a unique combination of five transgenic lines and each line was exclusively represented in two pools. Pooled libraries were sequenced using an Illumina MiSeq (Illumina) which produced 250 bp paired-end reads. Reads were aligned against the

biotinylated primer sequence, then unique sequences within each read subsequently aligned to a zebrafish reference genome (danRer10). Sequences common to all pools were assumed to be off-target and removed from analysis. Remaining reads from each pool were cross-referenced to the combination of embryos in each pool. Regions that had high and specific enrichment in both pools containing DNA from a particular sample were assigned as candidate insertion sites for that sample. To validate this procedure, we confirmed the map position for four lines through direct PCR genotyping.

## Expression analysis

To assess the selectivity of transgene expression, we manually set an intensity threshold for each line to distinguish cell body labeling from background, and calculated the proportion of voxels in the total cell body volume with a super-threshold signal. In assessing total brain coverage by the Cre library, we excluded *y457-Cre* which has extremely broad (possibly pan-neuronal) expression. To quantitatively describe transgene expression patterns, we created $20 \times 20 \times 20$ µm cubic bins (total 1804 bins) that were each entirely within the left hemisphere of the brain. For each line, we measured the mean expression within each bin and its corresponding volume on the right hemisphere. The resulting 1804 element vector was re-scaled in the range 0 to 1.0 to describe the relative intensity of transgene expression in each bin. *Supplementary file 3* reports these values together with the location of center voxel for each bin (Horizontal: dorsal to ventral; Transverse: Anterior to Posterior; Sagittal: Left to Right). In the downloadable version of ZBB2, this position can be recalled by using View → Jump to Frame, then entering the coordinates provided in first three columns in *Supplementary file 3* (e.g. 230, 90, 210 to access the center position for the first volume in the Table).

To estimate the total number of cells at six dpf, we dissected brains and counted dissociated cells using a cell sorter, yielding a total of $124700 \pm 2200$ cells per brain (mean and standard error, N = 9), including mature and immature neurons, glia and non-neural cells (e.g. connective tissue and vasculature).

For this procedure, brains were dissected at room temperature in 1x PBS (K-D Medical), dissociated in papain (Papain Dissociation System, Worthington Biochemical) by incubation in 20 U/mL papain for 20 min then triturated and 0.005% DNAse added. The final volume was adjusted to 200 µL in DMEM fluorobrite (Gibco) and immediately counted using a BD FACSCalibur (BD Biosciences), using a custom gate to include single cells in the forward and side scattered area. To minimize cell death, the entire process was completed within 45 min per brain.

We estimated the number of differentiated neurons by manually counting fluorescently labeled post-mitotic neurons in brain images. For this, we first imaged *huC:h2b-GCaMP6* transgenic brains at high-resolution ($0.5 \times 0.5 \times 0.5$ µm per voxel). In this line, nuclear-localized GCaMP fluorescence provides a discrete signal that facilitates identification of single neurons even in dense regions. We assigned all voxels in the brain to five neuronal-density bins based on the ratio of *huC:nls-mCar* (somas), and *huC:Gal4, UAS:syp-RFP* (synapses; see *Figure 4A*) and identified five representative volumes ($30 \times 30 \times 30$ µm each) for each of the five bins. We then counted neurons in *huC:h2b-GCaMP6* brain scans in each of the 25 volumes. Finally, we scaled the mean number of neurons in each bin by the relative fraction of the brain that each bin covers to obtain an estimate of the total neuron number. Using this procedure, we estimated that there are $92,000 \pm 3000$ mature neurons in the six dpf brain (N = 3 larvae).

To analyze the regional distribution of cell types based on molecular marker expression, we calculated the mean transgene expression intensity for 72 manually annotated anatomical structures (Z-Brain, *Supplementary file 4*) and for 168 computational defined brain regions (subset of Pajevic 180 regions, excluding narrow regions on the perimeter of the brain, *Supplementary file 5*) (*Gupta et al., 2018*; *Randlett et al., 2015*). For each line analyzed, voxel expression values were scaled 0–1, and means thus reflect the relative intensity of expression for regions within each maker and can not be compared between markers.

## Zebrafish Brain Browser software

The lines scanned and registered here were incorporated into the locally run Zebrafish Brain Browser, which requires downloading and installing the free IDL runtime environment. ZBB2

(including software and full resolution datasets) can be downloaded from our website (https://science.nichd.nih.gov/confluence/display/burgess/Brain+Browser).

To increase accessibility we also implemented an online version of ZBB2 that does not require downloading, and runs in any javascript-enabled web-browser (http://zbbrowser.com). We used Bootstrap (http://getbootstrap.com/) for interface design and jQuery for event-handling (https://jquery.com/). For rendering of 2D slices and 3D projections, we used X3DOM, a powerful set of open-source 3D graphics libraries for web development which integrates the X3D file format into the HTML5 DOM (*Behr et al., 2009*; *Congote, 2012*; *John et al., 2008*; *Polys and Wood, 2012*). ZBB2 uses X3DOM's built in *MPRVolumeStyle* and *BoundaryEnhancementVolumeStyle* functions to render 2D image files (texture atlases) in 3D space. The *MPRVolumeStyle* is used for the X, Y and Z slicer views to display a single slice from a 3D volume along a defined axis. We modified X3DOM source code for this volume style to support additional features including color selection, contrast and brightness controls, rendering of crosshairs, spatial search boxes and intersections between selected lines. The *BoundaryEnhancementVolumeStyle* renders the 3D projection. We also modified this function's source code, including additions of color, contrast, and brightness values. Other minor changes were made to the X3DOM libraries including a hardcoded override to allow additive blending of line colors. The online ZBB2 loads images of each line as a single 2D texture atlas. Image volumes for each line were converted to a montage, downsampling by taking every 4th plane in the z-dimension, and to 0.25, 0.5, and 0.75 their original size for low, medium, and high resolutions respectively to ensure rapid loading time. Texture atlas images were then referenced using X3DOM's 'ImageTextureAtlas' node, and its 'numberOfSlices', 'slicesOverX', and 'slicesOverY' attributes were specified as 100, 10, and 10, respectively. These atlases were then referenced by 'VolumeData' nodes, along with an *MPRVolumeStyle* or *BoundaryEnhancementVolumeStyle* node, to build the volumes visible on the screen.

To implement the 3D spatial search in the online edition of ZBB2, we first binarized and 4x-downsampled the resolution of each line. The data for each line was then parsed into a single array (width, height, depth). We compressed adjacent binary values into a single byte using bit shifting operators, downsampling the data once again by eight times. While greatly downsized, the entire dataset was still much too large to quickly download. We therefore fragmented the array for each line into 8 × 8×8 blocks of 64 bytes each, and concatenated blocks for every line, creating a single array of around 17 kb for a specific sub-volume of the brain. After the user defines the search volume, relevant volume fragments are downloaded and searched. Data from each fragment file is passed to a JavaScript Web Worker, allowing each file to be searched in a separate thread. This procedure facilitates minimal search times, with the main limitation being that thousands of binary files must be regenerated whenever a new line is added to the library.

## Quantification and statistical analysis

Analysis was performed with IDL (http://www.harrisgeospatial.com/SoftwareTechnology/IDL.aspx), Gnumeric (http://projects.gnome.org/gnumeric/) and Matlab (Mathworks).

## Resource sharing

Most enhancer trap lines are available from Zebrafish International Resource Center (https://zebrafish.org), with all others available from the authors upon request. Registered individual confocal brain scans can be downloaded from Dryad (https://doi.org/10.5061/dryad.tk467n8). Brain browser javascript code can be downloaded from GitHub (*Hurt et al., 2018*; copy archived at https://github.com/elifesciences-publications/ZebrafishBrainBrowser).

## Acknowledgments

We thank Steven Coon and James Iben from the NICHD Molecular Genomics Core and Lisa Williams-Simons from the NICHD FACS core for vital assistance. We are grateful to Jeremy Swan for help with interface design. This study utilized the high-performance computational capabilities of the Biowulf Linux cluster at the National Institutes of Health, Bethesda, MD (https://hpc.nih.gov/). The authors acknowledge Advanced Research Computing at Virginia Tech for providing computational resources and technical support that have contributed to the results reported within this paper

([http://www.arc.vt.edu](http://www.arc.vt.edu)). This work was supported by the Intramural Research Program of the *Eunice Kennedy Shriver* National Institute for Child Health and Human Development.

## Additional information

### Funding

| Funder | Grant reference number | Author |
|---|---|---|
| Eunice Kennedy Shriver National Institute of Child Health and Human Development | 1ZIAHD008884-04 | Harold A Burgess |
| Virginia Tech | Advanced Research Computing | Nicholas F Polys |

The funders had no role in study design, data collection and interpretation, or the decision to submit the work for publication.

### Author contributions

Kathryn M Tabor, Conceptualization, Data curation, Formal analysis, Validation, Investigation, Methodology, Writing—original draft, Writing—review and editing; Gregory D Marquart, Data curation, Investigation, Visualization, Methodology; Christopher Hurt, Software, Writing—original draft; Trevor S Smith, Data curation, Formal analysis, Investigation, Writing—review and editing; Alexandra K Geoca, Data curation, Investigation; Ashwin A Bhandiwad, Formal analysis, Investigation, Methodology; Abhignya Subedi, Formal analysis, Investigation; Jennifer L Sinclair, Hannah M Rose, Investigation; Nicholas F Polys, Conceptualization, Resources, Software, Supervision; Harold A Burgess, Conceptualization, Data curation, Software, Formal analysis, Supervision, Funding acquisition, Validation, Investigation, Methodology, Writing—original draft, Writing—review and editing

### Author ORCIDs

Kathryn M Tabor [http://orcid.org/0000-0003-3696-4584](http://orcid.org/0000-0003-3696-4584)
Gregory D Marquart [http://orcid.org/0000-0001-9811-5372](http://orcid.org/0000-0001-9811-5372)
Harold A Burgess [http://orcid.org/0000-0003-1966-7801](http://orcid.org/0000-0003-1966-7801)

### Ethics

Animal experimentation: This study was performed in strict accordance with the recommendations in the Guide for the Care and Use of Laboratory Animals of the National Institutes of Health. All of the animals were handled according to approved institutional animal care and use committee (IACUC) protocols (#15-039) of the Eunice Kennedy Shriver National Institute of Child Health and Human Development.

### Decision letter and Author response

Decision letter [https://doi.org/10.7554/eLife.42687.018](https://doi.org/10.7554/eLife.42687.018)
Author response [https://doi.org/10.7554/eLife.42687.019](https://doi.org/10.7554/eLife.42687.019)

## Additional files

### Supplementary files

• Supplementary file 1. Summary of Cre lines generated by our lab, available in ZBB2. Integration site given in zv10 coordinates, transgene orientation indicated in brackets.
Integration sites inside genes are annotated, or nearest Refseq gene within ~50 kb indicated.
DOI: [https://doi.org/10.7554/eLife.42687.009](https://doi.org/10.7554/eLife.42687.009)

• Supplementary file 2. Summary of all Gal4 lines generated by our lab, available in ZBB2. Coordinate system for mapped lines is either zv9 or zv10 as indicated.
DOI: [https://doi.org/10.7554/eLife.42687.010](https://doi.org/10.7554/eLife.42687.010)

• Supplementary file 3. Transgene expression intensity in small volumes within the brain. First three columns provide central coordinates of 20 × 20 × 20 μm volumes in the ZBB reference volume (Horizontal plane: distance in microns from the dorsal surface of the volume; Transverse plane: distance from the anterior edge of the volume; Sagittal plane: distance from the left side of the volume). Additional columns indicate the mean expression per bin of each transgene after scaling voxels from 0 to 1.0.
DOI: https://doi.org/10.7554/eLife.42687.011

• Supplementary file 4. Cell-type expression levels in neuroanatomical regions. Mean expression levels of transgenic markers in neuroanatomical regions defined by Z-Brain. Cell type markers: Glutamatergic (*vglut2a:DsRed*), GABAergic (*gad1b:GFP*), Glycinergic (*glyt2:GFP*), Catecholaminergic (*vmat2: GFP*), Serotonergic, (*pet1:GFP*), Cholinergic (*chata:Gal4*), Dopaminergic (*th:Gal4*), Hypocretin (*hcrt: RFP*), and glia (*gfap:GFP*).
DOI: https://doi.org/10.7554/eLife.42687.012

• Supplementary file 5. Cell-type expression levels in neuroanatomical regions. Mean expression levels of transgenic markers in computationally defined neuroanatomical regions.
DOI: https://doi.org/10.7554/eLife.42687.013

• Transparent reporting form
DOI: https://doi.org/10.7554/eLife.42687.014

## Data availability

Registered individual confocal brain scans have been deposited in Dryad https://doi.org/10.5061/dryad.tk467n8. Brain browser javascript code can be downloaded from https://github.com/BurgessLab/ZebrafishBrainBrowser; copy archived at https://github.com/elifesciences-publications/ZebrafishBrainBrowser.

The following dataset was generated:

| Author(s) | Year | Dataset title | Dataset URL | Database and Identifier |
|---|---|---|---|---|
| Tabor KM, Marquart GD, Smith TS, Geoca A, Bhandiwad A, Hannah M Rose, Jennifer L Sinclair | 2018 | Brain-wide cellular resolution imaging of Cre transgenic zebrafish lines for functional circuit-mapping | http://dx.doi.org/10.5061/dryad.tk467n8 | Dryad Digital Repository, 10.5061/dryad.tk467n8 |

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
