## [Decision Letter]

Thank you for submitting your article "Brain-wide cellular resolution imaging of Cre transgenic zebrafish lines for functional circuit-mapping" for consideration by *eLife*. Your article has been reviewed by three peer reviewers, and the evaluation has been overseen by a Reviewing Editor and Didier Stainier as the Senior Editor. The following individuals involved in review of your submission have agreed to reveal their identity: Claire Wyart (Reviewer #1). The other reviewers remain anonymous.

The reviewers have discussed the reviews with one another and the Reviewing Editor has drafted this decision to help you prepare a revised submission.

Summary:

This tool/resource provides a large set of zebrafish Cre and Gal4 lines for use by the zebrafish research community for the generation of specific lines through intersectional genetics. The data are organized into an online platform that makes the content readily accessible to investigators.

Essential revisions:

The reviewers expressed enthusiasm about the platform as well as the care that went into the work. These comments are listed below under "general assessments" from each reviewer. Two points of concern/interest were raised.

1) Reviewer 3 brought up the point that it is not clear whether the intersectional approach will work well or not with fish in which both Gal4 and cCre are expressed in only a small number of neurons. The idea here is that it should, but evidence is not provided that it does. (Reasons for questioning whether it does are given in the reviewer's comments, but include stochastic activity at target loxP sites for Cre, and variable expression and silencing for Gal4.) In the consultation, the other reviewers agreed with reviewer 3's concern, and therefore suggest that an example or two be provided. We realize this will require some breeding time if no such data is already in hand; please let us know if this is feasible.

2) Reviewer 1 suggested that the utility of the resource could be greatly increased by a somewhat deeper analysis of the existing data, e.g., by providing information on neurotransmitter type by quantifying overlap with transgenic lines targeting specific neurotransmitter types. This would not require new data, but additional information on the existing data.

These points are more fully spelled out in the collected major comments from the reviewers, which are listed below, with editorial indications in square brackets.

General assessments:

Reviewer #1:

The manuscript of Tabor et al. provide a very important set of cre and gal4 lines that will be extremely useful for intersectional genetics in the field of zebrafish neuronal circuits, together with a novel online platform, which is extremely convenient to analyse the profile of expression online using ZBB2. I believe that this effort is determinant for the expansion of the field of the investigation of neuronal circuits underlying behavior in zebrafish larvae.

Reviewer #2:

This paper is a resources and tools contribution that presents many (100ish) new zebrafish CRE and gal4 lines to facilitate intersectional genetic strategies in fish. Importantly, the lines are presented in a beautiful online (or downloadable) browser that makes it very easy to explore them to decide which ones are useful, while learning about the larval fish brain at the same time. Accessibility and easy evaluation of the lines is as important as the lines themselves. The combination of lines with the ability to easily explore them and their relationships and intersection with other lines makes for a strong contribution. I have no substantive issues with a well presented and very useful tool. All the materials and fish are made accessible.

Reviewer #3:

In this manuscript Tabor et al. have generated more than 50 enhancer trap lines that express Cre in defined population of cells in the CNS. The authors obtained high-resolution images of these new lines along with 45 Gal4 enhancer trap lines that have not yet been reported. The authors then registered these images with the Zebrafish Brain Browser (ZBB) atlas, software that was created by the authors in a previous study (Marquart et al., 2015). Together with the images that have already been reported (109 lines), the new database (ZBB2) now include 65 Cre lines and 158 Gal4 lines. Additionally, the authors provide an online interface to the atlas. There is no doubt that these resources will be very useful for the zebrafish neuroscience community.

Major comments [concatenated and reordered, from all reviewers]:

1) [Related to Essential revision 1] Throughout the manuscript, the words, "cellular resolution", "single cell resolution", and "intersectional targeting" are used multiple times. With these words, readers would expect the following experiments: searching for Cre and Gal4 lines in the database, finding small numbers of neurons in which Cre and Gal4 expression overlap, and expressing reporter/effecter genes in these neurons. Nowhere in the manuscript, however, are examples of such experiments are provided. Thus, it is not clear whether targeting expression of reporter genes with cellular resolution is possible with the combination of Cre and Gal4. There are several potential concerns for the intersectional targeting of small numbers of neurons. These include stochastic activity at target loxP sites for Cre, and variable expression and silencing for Gal4. The authors cite the Current Biology paper (Tabor et al., 2018) as an example of intersectional targeting, but the Cre lines used in that study are broad expression lines (Cre expression in particular rhombomeres). So, the study does not serve as an example of intersectional targeting at cellular resolution. There is one place in the paper where the authors mention UAS:Switch expression experiments (in Discussion). However, it is written in a negative context: "Occasionally, we have observed UAS:Switch expression in neurons outside the domain of Cre expression in the ZBB2 atlas". In any case, the authors need to show clear examples of intersectional targeting in small numbers of neurons where Cre and Gal4 expression overlap in the ZBB2database.

2) [Related to Essential revision 2] While I do not request more experiments to the authors, I think an in-depth analysis of the distribution of interneurons types based on neurotransmitter and neuromodulator expressions per brain area, which seems amenable to the authors based on the data acquired, would provide more relevant information to the reader and the field. In particular, it would be useful to take it a step further and analyze out of the ~92 000 neurons in the brain the putative distribution as a function of neurotransmitter, neuromodulator and/or neuropeptide types. This effort would constitute a critical resource for the field and seems possible to estimate with ZBB2 and the collection of transgenic lines mapped in the 6 dpf zebrafish larval brain.

3) [Related to Essential revision 2] Related to Figure 3: Seeing images of the lines or 3D views online is very nice qualitatively. Naive question here: I wonder if we could take it one step further to reach a quantitative representation of the expression over the entire brain space of the larva. Do the authors have a simple way to converge on a physical 3D map indexed in a matrix with absolute coordinates that we could refer to in publications from different labs?

4) [Related to Essential revision 2] In order to locate with absolute coordinates the expression of transgenes better than by eye on the atlas: could the authors come up with a matrix of ~1000 bins (containing on average 10 neurons) for which each index corresponds to a graded value of expression on a 0-1 scale?

---

## [Author Response]

Major comments [concatenated and reordered, from all reviewers]:1) [Related to Essential revision 1] Throughout the manuscript, the words, "cellular resolution", "single cell resolution", and "intersectional targeting" are used multiple times. With these words, readers would expect the following experiments: searching for Cre and Gal4 lines in the database, finding small numbers of neurons in which Cre and Gal4 expression overlap, and expressing reporter/effecter genes in these neurons. Nowhere in the manuscript, however, are examples of such experiments are provided. Thus, it is not clear whether targeting expression of reporter genes with cellular resolution is possible with the combination of Cre and Gal4. There are several potential concerns for the intersectional targeting of small numbers of neurons. These include stochastic activity at target loxP sites for Cre, and variable expression and silencing for Gal4. The authors cite the Current Biology paper (Tabor et al., 2018) as an example of intersectional targeting, but the Cre lines used in that study are broad expression lines (Cre expression in particular rhombomeres). So, the study does not serve as an example of intersectional targeting at cellular resolution. There is one place in the paper where the authors mention UAS:Switch expression experiments (in Discussion). However, it is written in a negative context: "Occasionally, we have observed UAS:Switch expression in neurons outside the domain of Cre expression in the ZBB2 atlas". In any case, the authors need to show clear examples of intersectional targeting in small numbers of neurons where Cre and Gal4 expression overlap in the ZBB2database.

In the manuscript we have generally used the phrases like 'cellular resolution' to refer to the level of detail observable in the database rather than the level of resolution that we think can be usually obtained by intersectional targeting with transgenic reagents. Rather, our vision is that Cre lines with relatively broad expression can be used to select out sub-domains within Gal4 expression patterns. To achieve true single cell resolution, a third intersectional reagent is generally necessary (for example, the B3 recombinase that we used in Tabor et al., 2018).

However, the reviewers correctly observe, in Tabor et al., we performed intersectional targeting using Gal4 and Cre lines with relatively broad expression. We also agree that UAS lines (including the UAS:Switch) tend to silence and show variable expression. Therefore in the revised manuscript we have added:

A) A new Figure 5 which illustrates examples where Gal4 and Cre intersectionally drive expression in a small groups of neurons. In the accompanying text, we explicitly acknowledge the confound of variable expression.

B) A short description to the Materials and methods section describing the protocol that we use to minimize transgene silencing.

2) [Related to Essential revision 2] While I do not request more experiments to the authors, I think an in-depth analysis of the distribution of interneurons types based on neurotransmitter and neuromodulator expressions per brain area, which seems amenable to the authors based on the data acquired, would provide more relevant information to the reader and the field. In particular, it would be useful to take it a step further and analyze out of the ~92 000 neurons in the brain the putative distribution as a function of neurotransmitter, neuromodulator and/or neuropeptide types. This effort would constitute a critical resource for the field and seems possible to estimate with ZBB2 and the collection of transgenic lines mapped in the 6 dpf zebrafish larval brain.

As requested, we provide new supplementary information (Supplementary files 4 and 5) that describe the density of neurotransmitter/neuromodulator expression in different brain regions. For this, we have used both manually curated anatomical masks from Z-Brain, and second set of computational defined spatially smaller masks recently described by our group (Gupta et al., 2018). However, we cannot reliably extend this to a neuron-level description (if we understand the idea correctly) due to biological variability. Almost all brain regions contain multiple cell types that are intermingled, and – with the possible exception of the Mauthner neuron – specific cells are not precisely located at the same coordinates in different individuals.

3) [Related to Essential revision 2] Related to Figure 3: Seeing images of the lines or 3D views online is very nice qualitatively. Naive question here: I wonder if we could take it one step further to reach a quantitative representation of the expression over the entire brain space of the larva. Do the authors have a simple way to converge on a physical 3D map indexed in a matrix with absolute coordinates that we could refer to in publications from different labs?

The ZBB reference volume constitutes a 3D map with an absolute coordinate system in microns. We modified the code for the downloadable version so that irrespective of user-selected downscaling values, slices are now labeled by their position in ZBB reference space. The online version already displays the absolute coordinates of the selected voxel. This position can be reported in publications.

4) [Related to Essential revision 2] In order to locate with absolute coordinates the expression of transgenes better than by eye on the atlas: could the authors come up with a matrix of ~1000 bins (containing on average 10 neurons) for which each index corresponds to a graded value of expression on a 0-1 scale?

This is a great idea – thank you. We have added new Supplementary file 3 that provides coordinates for 1804 bins (20 µm side volumes) and the density of transgene expression in each. Using the bin coordinates, users can quickly navigate to the selected region in the atlas.